# Enzymatic Antioxidant Defense System of Scots Pine Seedlings Under Conditions of Progressive Manganese Deficiency

**DOI:** 10.3390/biology15010101

**Published:** 2026-01-04

**Authors:** Yury V. Ivanov, Alexandra I. Ivanova, Alexander V. Kartashov, Galina V. Glushko, Polina P. Loginova, Vladimir V. Kuznetsov

**Affiliations:** K.A. Timiryazev Institute of Plant Physiology, Russian Academy of Sciences, Botanicheskaya Street 35, 127276 Moscow, Russia; aicheremisina@mail.ru (A.I.I.); botanius@yandex.ru (A.V.K.); g.glushko-v@yandex.ru (G.V.G.); pptatarkina@mail.ru (P.P.L.)

**Keywords:** *Pinus sylvestris*, superoxide dismutase, native PAGE, guaiacol peroxidase, catalase, ascorbate peroxidase

## Abstract

Manganese (Mn) is a vital micronutrient for plants. In particular, Mn plays an important role in the plant life cycle, participating in processes such as photosynthesis, respiration, reactive oxygen species (ROS) scavenging, pathogen defense, and hormonal signaling. Currently, approximately 400 enzymes, including the enzymes of Photosystem II, are known to contain Mn in the metal-binding site. Impaired photosynthesis leads to the formation of ROS, which can damage DNA, inactivate proteins, and impair cell membranes. It is generally believed that Mn deficiency leads to the generation of ROS. Plants typically respond to increased ROS levels by increasing the activity of antioxidant enzymes to neutralize excess oxygen radicals. Unlike most plants, pine seedlings, as shown in our study, respond to progressive Mn deficiency in a completely different way. Specifically, a twofold decrease in the activity of one of the key antioxidant enzymes, superoxide dismutase (SOD), which is present in Cu/Zn-containing forms, was observed in the needles of Mn-deficient plants. The activities of other components of the antioxidant defense system remained unchanged or, as in the case of peroxidase, even decreased. This paper discusses possible causes for the paradoxical response of pine seedlings to Mn deficiency.

## 1. Introduction

Manganese (Mn) plays a crucial role in the plant life cycle, participating in processes such as photosynthesis, respiration, reactive oxygen species (ROS) scavenging, pathogen defense, and hormonal signaling [1,2,3]. Currently, approximately 400 enzymes containing Mn at the metal-binding site are known, but only 20% of them have experimental data supporting the use of Mn as a cofactor [1,2]. In many enzymes, Mn is thought to be interchangeable with other divalent cations, such as magnesium, calcium, zinc, copper, and cobalt. However, the oxygen-evolving complexes of Photosystem II (PSII), manganese superoxide dismutase (Mn-SOD) and oxalate oxidase specifically require Mn for their function [1,2].

Owing to its essential role in PSII, Mn deficiency has negative effects on the photosynthetic apparatus. Impaired photosynthetic electron transport leads to the accumulation of ROS and the development of oxidative stress [2,4,5]. ROS formation in redox reactions is inevitable during natural plant metabolism, but under optimal conditions, their formation and utilization are tightly controlled in plants [1,6,7]. SODs, considered the first line of defense against oxidative stress, catalyze the disproportionation of superoxide into molecular oxygen and hydrogen peroxide [1,7]. Hydrogen peroxide is utilized by catalases (CAT) and peroxidases [6]. The activity of ascorbate peroxidase (APX), which is localized in chloroplasts, is particularly important in counteracting the development of oxidative stress [6].

One of the visual signs of acute Mn deficiency in plants is the appearance of necrotic spots on leaves [4], presumably due to increased ROS production and decreased SOD activity, especially of Mn-containing isozymes [1,2]. However, the utilization of the cellular Mn pool under low Mn conditions varies greatly among photosynthetic organisms [2]. Moreover, in higher plants, Mn-SOD is not widely distributed and is localized mainly in mitochondria and peroxisomes [8], indicating that it is not directly involved in the detoxification of ROS generated in chloroplasts. Nevertheless, the changes in the activity of antioxidant enzymes under Mn deficiency conditions are very diverse. For example, in the leaves of Mn-deficient tobacco plants, the activities of total SOD, Mn-SOD + Fe-SOD, Cu/Zn-SOD, and APX were substantially decreased [9], whereas the leaves of Mn-deficient soybean (*Glycine max* L. Merr.) plants were characterized by increased CAT and peroxidase activities [10]. Mn deficiency increases the activity of cytosolic Cu/Zn-SOD and mitochondrial Mn-SOD in tomato leaves [11] but does not result in a loss of Mn-SOD activity in *Arabidopsis* plants [12]. In Mn-deficient mulberry (*Morus alba*) plants, increased lipid peroxidation was observed alongside increased total SOD and catalase activities [13]. In contrast, in Norway spruce, Mn deficiency did not result in increased lipid peroxidation in needles despite increased SOD activity [14].

In a previous study, we demonstrated that Scots pine (*Pinus sylvestris* L.) seedlings subjected to manganese deficiency experienced suppressed growth and photoinhibition of PSII but did not experience signs of oxidative stress [15]. However, the function of the enzymatic antioxidant defense system under these conditions remains unclear.

The purpose of this study was to examine the changes in SOD, APX, CAT, and guaiacol peroxidase (POD) activities in Scots pine seedling organs over an extended period (up to 24 weeks) of Mn deficiency. Our study was based on two key assumptions:Manganese deficiency per se should not lead to a decrease in total SOD activity, as the activity of Mn-containing SOD isozymes in Scots pine accounts for only 1–4% of total SOD activity [16].The absence of oxidative stress in plant organs under Mn deficiency conditions, determined by the lack of an increase in the contents of malondialdehyde and 4-hydroxyalkenals in the roots and needles [15] may be attributed to a coordinated increase in the activities of antioxidant enzymes in response to these conditions.

## 2. Materials and Methods

### 2.1. Experimental Design

Scots pine (*Pinus sylvestris* L.) sprouts were grown from seeds in polypropylene cartridges filled with agar in 6 L plastic trays in distilled water. After the roots emerged through the agar layer, the plants were divided into two groups: control (with added Mn) and Mn-deficient (without added Mn). The seedlings were cultivated for 24 weeks in a growth chamber under fluorescent lighting at 130 ± 15 μmol m^−2^ s^−1^, with a 16 h photoperiod and a constant air temperature of 24 ± 2 °C. The growth medium used had the following composition: 2.0 mM NH_4_NO_3_, 1.5 mM KH_2_PO_4_, 1.0 mM CaCl_2_, 0.5 mM MgSO_4_, 0.1 mM Na_2_SO_4_, 55 μM H_3_BO_3_, 5 μM MnSO_4_, 1.26 μM ZnSO_4_, 0.32 μM CuSO_4_, 0.1 μM Na_2_MoO_4_, 0.02 μM Co(NO_3_)_2_, 1.0 μM KI, 9.5 μM FeSO_4_, and 9.5 μM Na_2_-EDTA at a pH of 4.5 [17]. For growing Mn-deficient plants, the same growth medium was used, but without Mn. The growth medium was continuously aerated and changed twice a week. Analysis of the Mn content in the growth medium revealed the following actual concentrations of Mn: 5.21 ± 0.33 μM for the control and 26.3 ± 2.2 nM for the Mn-deficient group [15].

The plant samples were collected at 6, 19, and 24 weeks after seed germination during the stages of intensive growth of juvenile needles, the beginning of perennial needle growth, and the formation of the first terminal bud, respectively. At the 6th week, 3–5 plants were collected and grouped into combined samples to separate the roots and needles. It was necessary to obtain an average sample weight of 150–200 mg. As the plants grew during the experiment, at the 19th and 24th weeks, two seedlings were combined, or individual seedlings were used. Similar to the process at the 6th week, the roots and needles were separated from the plants. All samples were fixed in liquid nitrogen and stored at −70 °C until biochemical analysis. Each combined sample was treated as a biological replicate [15].

### 2.2. Determining the Contents of Manganese

To determine the Mn ion content in the organs, the plant roots were washed with a 20 mM aqueous solution of Na_2_-EDTA to remove ions adsorbed on the surface. The seedlings were then separated into roots and needles and dried until a constant weight was reached. The samples were then digested in solutions of concentrated HNO_3_ and HClO_4_ (2:1 (*v*/*v*)), after which the Mn content was determined by atomic absorption spectrometry [15,17].

### 2.3. Determining of Fresh and Dry Weights and Calculation of Dry Matter Content

To estimate the fresh weights of the needles and roots, an analytical balance with an accuracy of 1 mg (Scout Pro SPU123, Ohaus Corporation, Parsippany, NJ, USA) was used. To determine the dry weights of the needles and roots, an analytical balance with an accuracy of 0.1 mg (AB54-S, Mettler Toledo, Greifensee, Switzerland) was used. Before analysis, the plant material was dried in a thermostat oven at 70 °C for three days [17].

The dry matter contents of the roots or needles were calculated as a percentage of the dry weight to the fresh weight of the corresponding sample [18].

### 2.4. Determining the Activities of Antioxidant Enzymes

A weighed sample of plant material of approximately 100–150 mg was ground in a porcelain mortar in liquid nitrogen and extracted with 1.0 mL of ice-cold 100 mM carbonate–bicarbonate buffer (pH = 10.3) containing 100 mg of suspended insoluble poly(vinylpolypyrrolidone) (CAS No. 9003-39-8), 1.0 mM DL-dithiothreitol (CAS No. 3483-12-3), and 0.5 mM phenylmethyl sulfonyl fluoride (CAS No. 329-98-6). The homogenate was centrifuged at 12,000× *g* for 10 min at 4 °C, and the resulting supernatant was used for determining enzyme activities [19,20]. All steps in the preparation of the enzyme extracts were carried out at 4 °C.

Total SOD activity was determined via a modification of the Beauchamp and Fridovich method [21]. The 2 mL reaction mixture contained 1.6 mL of 50 mM Tris-HCl buffer, pH 7.8, 0.05 mL of supernatant, 0.2 mL of 0.1 M DL-methionine (CAS No. 59-51-8), 0.05 mL of 2 mM nitrotetrazolium blue chloride (CAS No. 298-83-9), 0.05 mL of 1% Triton^TM^ X-100 (CAS No. 9036-19-5) and 0.05 mL of 0.005% riboflavin (CAS No. 83-88-5). The reaction was carried out under illumination with white LEDs of 500 ± 50 μmol m^−2^ s^−1^ for 15 min. The absorbances of the samples were measured at 560 nm with a Genesys 10 UV–Vis spectrophotometer (Thermo Fisher Scientific, Waltham, MA, USA) against a control that did not contain sample extract.

The CAT activity was assayed by measuring the initial rate of hydrogen peroxide disappearance via the Chance and Maehly method [22] with the following modifications. The 2.0 mL reaction mixture contained 1.85 mL of 0.05 M K/Na-phosphate buffer, pH 7.8, 0.05 mL of supernatant, and 0.1 mL of 0.1 M H_2_O_2_. The dynamics of the changes in optical density were recorded on a spectrophotometer for 3 min at a wavelength of 240 nm. Measurements were taken every 5 s. To calculate the enzyme activity, a calibration curve with hydrogen peroxide concentrations ranging from 0.5 to 10 mM was used.

The POD activity was measured as the oxidation of guaiacol [22,23] with the following modifications. The 2.45 mL reaction mixture contained 1.95 mL of 0.066 M K/Na-phosphate buffer, pH 7.4; 0.05 mL of supernatant; 0.2 mL of 20 mM guaiacol (CAS No. 90-05-1); and 0.25 mL of 50 mM H_2_O_2_. The dynamics of the changes in optical density were recorded on a spectrophotometer for 3 min at a wavelength of 470 nm. Measurements were taken every 5 s. POD activity was expressed as µmol guaiacol per mg soluble protein per minute.

The APX activity was determined spectrophotometrically according to the rate of destruction of ascorbic acid via the method of Nakano and Asada [24]. The 2.0 mL reaction mixture contained 1.90 mL of 0.5 mM ascorbic acid and 0.1 mM Na_2_-EDTA in 0.05 M K/Na-phosphate buffer, pH 7.0, 0.05 mL of supernatant, and 0.05 mL of 0.1 mM H_2_O_2_. The dynamics of the changes in optical density were recorded on a spectrophotometer for 3 min at a wavelength of 290 nm. Measurements were taken every 5 s.

Aliquots of the extracts were used to determine the soluble protein content of the samples in accordance with Esen’s method [25]. Bovine serum albumin (CAS No. 9048-46-8) was used to construct the calibration curve.

### 2.5. Native Polyacrylamide Gel Electrophoresis (PAGE) of SOD

Enzyme extracts for native PAGE were obtained from the needles of 24-week-old plants via the method described above, but the volume of the extraction buffer was reduced to 800 µL. Electrophoresis of the protein fraction was performed in a 10% polyacrylamide gel without added SDS [26] in Tris-glycine buffer at a concentration of 25 mM Tris and 192 mM glycine in a Mini-PROTEAN^®^ 3 cell electrophoresis chamber (Bio-Rad Laboratories, Inc., Hercules, CA, USA) at 150 V and 4 °C. Electrophoresis lasted 6.5 h, including 3 h after bromophenol blue (CAS No. 115-39-9) was released from the gel. Enzyme samples loaded into loading wells were matched for soluble protein content. Preliminary experiments revealed that the best separation of SOD isozymes was achieved with a loading of 40 μg of protein (Appendix A). The soluble protein contents of the enzyme extracts were determined via the Bicinchoninic Acid Kit for Protein Determination (Sigma-Aldrich, St. Louis, MO, USA, BCA1) according to the manufacturer’s instructions.

To visualize the SOD isozymes after electrophoresis, the gel was stained for 30 min in the dark in 50 mM Tris-HCl buffer (pH 7.8) containing 150 μM riboflavin (CAS No. 83-88-5) and 250 μM nitroblue tetrazolium (CAS No. 298-83-9). The polyacrylamide gel was then thoroughly washed with distilled water and placed under LEDs (100 ± 10 μmol m^−2^ s^−1^) for 9 min until SOD isozymes became visible [21]. To test the performance of the selected system, a commercial SOD from bovine erythrocytes (BioChemika, Espoo, Finland, 86200) was used. To selectively inhibit SOD isozymes, gels were incubated for 30 min before being stained in solutions of 3 mM KCN (inhibition of Cu/Zn-SOD), 5 mM H_2_O_2_ (inhibition of Fe-SOD and Cu/Zn-SOD) [6,16], or 10 mM NaN_3_ (inhibition of Fe-SOD and Mn-SOD) [27].

After visualization, the polyacrylamide gels were individually laid out on the glass of an Epson Perfection V500 Photo flatbed scanner (Epson, Japan) and scanned at a resolution of 600 dpi.

The quantitative assessment of SOD isozyme activity (based on the signal level of the corresponding spot) was performed via Image Studio Digits V5.2 (LI-COR, Inc., Lincoln, NE, USA). The brightness signal values were normalized by subtracting the background values (the arithmetic mean of the background signal in a 3-pixel-wide region around the analyzed spot). The resulting signal intensity for the spot corresponding to the SOD isozyme was expressed in relative units.

For additional analysis of SOD activity, crude enzyme extracts were purified on desalting columns with Sephadex G-25 resin (PD MiniTrap™ G-25, GE HealthCare, Chicago, IL, USA) according to the manufacturer’s instructions.

### 2.6. Statistical Analysis

Six to twelve biological replicates were used to determine the Mn content in the roots and needles, the dry weights of the roots and needles, and the dry matter contents in the roots and needles. Four to six biological replicates were used to determine the enzyme activity and soluble protein content in the seedling organs. Eight biological replicates were used to determine the SOD activity in needles via native PAGE. Statistical analyses of the data were performed via SigmaPlot 12.5 (Systat Software Inc., Chicago, IL, USA) with one-way analysis of variance (ANOVA) followed by Duncan’s post hoc test for normally distributed data (significant differences at *p* < 0.05 denoted by different regular letters), Kruskal–Wallis one-way ANOVA on ranks followed by the Student–Newman–Keuls post hoc test (significant differences at *p* < 0.05 denoted by different italic letters) or Dunn’s post hoc test (significant differences at *p* < 0.05 denoted by different bold italic letters) for non-normally distributed data and data with unequal variance. Pairwise comparisons of the means with controls at corresponding time points were performed using Student’s *t*-test for normally distributed data (significant differences at *p* < 0.05 denoted by asterisks (*)) or the Mann–Whitney rank sum test when the *t*-test was not applicable (significant differences at *p* < 0.05 denoted by multiplication symbols (×)) using SigmaPlot 12.5. The data presented in the figures and table are the arithmetic mean values ± standard errors.

## 3. Results

### 3.1. Manganese Content in Plant Organs

As previously noted [15], the Mn content in the roots of the control plants was highly variable, especially at the beginning of the experiment. Moreover, in the Mn-deficient plants, the root Mn content was significantly lower than that in the control plants and stabilized at 0.10 μmol/g DW beginning in the 19th week of the experiment. The needle Mn content in the control plants remained constant, whereas in the Mn-deficient plants, it decreased throughout the experiment. At week 6, the manganese content was 16.9 times lower than that in the control, and at week 24, it was 59.6 times lower (Table 1).

### 3.2. Effects of Mn Deficiency on Seedling Growth

The growth of Scots pine seedlings in an Mn-free nutrient solution for 6 weeks did not negatively affect plant biomass accumulation. In contrast, the dry weights of the needles (Figure 1A) and roots (Figure 1B) of the Mn-deficient plants were 19.6% and 33.8% greater than those of the control plants. However, by the 19th week of the experiment, the increase in needle and root weights in the control was 7.6- and 7.2-fold, respectively, whereas in the Mn-deficient plants it was only 4.1- and 3.3-fold, respectively. As a result, the growth of Mn-deficient plants was delayed by 34.8% for needle DW and by 38.2% for root DW (Figure 1). By the 24th week of the experiment, the delay in needle growth in the Mn-deficient plants compared with the control increased to 46.8% (Figure 1A). The DW of the roots of the Mn-deficient plants was 40.5% lower than that of the control plants (Figure 1B).

The dry matter content in the needles of the Mn-deficient plants at the 6th week of the experiment was 3.3% greater than that in the control plants. Subsequently, no statistically significant differences were detected between the plant groups, although the rate of dry matter accumulation in the needles of the Mn-deficient plants was slightly lower than that in the control plants. Thus, in the needles of the control plants, the dry matter content increased by 33.9% and 49.6% by the 19th and 24th weeks of the experiment, respectively. In the needles of Mn-deficient plants, this increase was 24.3% and 32.4%, respectively (Figure 1C). Unlike the needles, no differences in the dry matter content in the roots of the seedlings were observed throughout the experiment (Figure 1D).

### 3.3. Soluble Protein Content

Mn deficiency did not result in changes in the soluble protein content of either the needles or the roots of the seedlings compared with that of the control plants (Figure 2). The protein content in the needles of both plant groups remained constant until week 19 of the experiment. However, by week 24, the protein content in the needles decreased by 44.1% and 43.8% in the control and Mn-deficient plants, respectively (Figure 2A). Interestingly, the changes in protein content in the seedling needles during the experiment did not affect the protein content in the seedling roots (Figure 2B).

### 3.4. SOD Activity

At the 6th week of the experiment, the SOD activity in the needles of the Mn-deficient plants was half that of the control. At the 19th week, these differences were 2.6-fold, and at the 24th week, they were 2.2-fold lower than those of the control (Figure 3A). In both the control and Mn-deficient plants, no differences in SOD activity were detected between the 6th and 19th weeks of the experiment. However, at the 24th week of the experiment, the SOD activity in the needles increased compared with the enzyme activity in the 6th week—2.6-fold in the control and 2.4-fold in the Mn-deficient plants. For seedling roots, no statistically significant differences in SOD activity were detected between the experimental treatments or the observation time points (Figure 3B).

### 3.5. Catalase Activity

The catalase activity in the needles of the Mn-deficient plants remained at a level comparable to that of the control plants throughout the experiment (Figure 3C). Both treatments resulted in an increase in catalase activity throughout the experiment, peaking at week 24. Overall, catalase activity increased by 45.2% in the needles of the control plants and by 98.2% in the needles of the Mn-deficient plants during the experiment. Mn deficiency had no effect on catalase activity in the roots of the seedlings (Figure 3D).

### 3.6. Guaiacol Peroxidase Activity

Like SOD activity, guaiacol peroxidase activity in the needles of Mn-deficient plants was reduced by 23.2% and 23.0%, respectively, at weeks 6 and 19 of the experiment compared with that of the control (Figure 3E). Additionally, from the 6th to the 19th weeks, a similar increase in POD activity was observed in the needles of both plant groups: 63.3% in the control group and 63.7% with Mn deficiency. There were no significant changes in POD activity in needles from the 19th to the 24th week. Mn deficiency did not affect POD activity in seedling roots (Figure 3F). However, POD activity in plant roots at the 24th week was, on average, 14.8-fold greater than that in the needles.

### 3.7. Ascorbate Peroxidase Activity

Mn deficiency had no effect on APX activity in either needles (Figure 3G) or seedling roots (Figure 3H). There were no trends or significant changes in APX activity detected in either needles or roots throughout the experiment.

### 3.8. Native PAGE of SOD

After SOD was separated on a polyacrylamide gel, three clearly distinguishable SOD isozymes were detected in both the control and Mn-deficient plants. Fourth, a less mobile, subactive SOD isozyme was not detected in all the samples (regardless of treatment) and was detected only when the gel was overloaded with a protein that hindered the separation of the main SOD isozymes (Figure 4A and Appendix A). Treatment with KCN or H_2_O_2_ resulted in the disappearance of all the SOD isozymes on the polyacrylamide gel (Figure 4B and Appendix A), whereas treatment with sodium azide resulted in the retention of all the SOD isozymes (Appendix A).

Spectrophotometric analysis of SOD activity in the enzyme extracts during polyacrylamide gel loading confirmed a 2.0-fold difference in activity between the control and Mn-deficient needles (Appendix A). After 7 h (corresponding to electrophoresis and polyacrylamide gel staining), no statistically significant changes in SOD activity in the extracts compared with the initial activity were observed. However, the difference in activity between the control and Mn-deficient samples was 1.8-fold (Appendix A). Analysis of the activity of the SOD isozymes on a polyacrylamide gel (Figure 5A and Appendix A) revealed no statistically significant differences between the control and Mn-deficient groups (Figure 5B). Notably, the decreasing activity in Mn-deficient plants was characteristic of only isozymes No. 1 (15.5%, *p* = 0.130) and No. 2 (27.3%, *p* = 0.307) (Figure 5B). The activity of isozyme No. 3 did not change. Isozyme No. 1 had the highest activity, accounting for approximately 68% of the total SOD activity.

Comparative analysis of SOD activity in the crude and purified extracts from pine needles at the 24th week confirmed the persistence of differences in activity between the control and Mn-deficient plants (Appendix A).

## 4. Discussion

The Mn content in the needles of the Mn-deficient plants decreased throughout the experiment (Table 1). Despite the substantially lower Mn content in the organs of the Mn-deficient plants at the 6th week of the experiment, this amount was sufficient to support normal growth. This was evidenced by the increased root and needle weights and needle dry matter content of the Mn-deficient plants compared with those of the control plants (Figure 1). However, by week 19 of the experiment, along with a progressive decrease in the Mn content in the needles, strong growth inhibition was also observed, indicating Mn deficiency in the plants [1,2]. However, the growth inhibition of the Mn-deficient plants did not lead to a decrease in the dry matter content or soluble protein content in the organs compared with those of the control group of plants (Figure 2). The lack of effect of Mn deficiency on protein content was also characteristic of Norway spruce. In this species, the Mn content in the needles decreased to 0.07 μmol/g DW (47-fold lower than that in the control), leading to a significant decrease in the content of photosynthetic pigments [14]. Notably, the soluble protein content in the needles significantly decreased from the 19th to the 24th week in both groups of plants (Figure 2A) despite a continued increase in the needle dry matter content (Figure 1C). This decrease in soluble protein content is likely due to a natural ontogenetic feature associated with the intensive development of perennial needles, the formation of the first terminal bud [28], and a decrease in the proportion of juvenile needles. Thus, the soluble protein content in the organs of the Mn-deficient plants, which was comparable to that of the control, may indicate the absence of a direct effect of Mn deficiency on the rate of protein synthesis or the rate of proteolysis, as observed under other stress conditions [29].

Considering the lack of effect of Mn deficiency on the soluble protein content in seedling organs, the twofold lower level of SOD activity in the needles of Mn-deficient plants (Figure 3A) was extremely unexpected. This activity was already observed at the 6th week of the experiment, when no negative effect on plant growth was noted (Figure 1). The progressive decline in the Mn content in the needles (Table 1) did not lead to a further decrease in SOD activity (Figure 3A). This differs significantly from the response of Scots pine to chronic zinc toxicity [30] or copper toxicity [31], which do not cause changes in total SOD activity in seedling needles. Thus, a 17-fold decrease in the Mn content in the needles of the Mn-deficient plants compared with the control was sufficient to halve the SOD activity in the needles of the Mn-deficient plants. Interestingly, the highest level of SOD activity was observed at the 24th week of the experiment, when the soluble protein content in the needles significantly decreased (Figure 2A). Moreover, increasing Mn deficiency did not prevent a significant (2.8-fold) increase in SOD activity in the needles of the Mn-deficient plants compared to the enzyme activity at the 19th week of the experiment (Figure 3A). Thus, the lower level of SOD activity in the needles of the Mn-deficient plants than in those of the control plants was not associated with Mn deficiency per se but was rather due to the restructuring of plant metabolism under these conditions.

Notably, the changes in SOD activity in plants under Mn deficiency conditions are diverse. For example, in Mn-deficient tobacco plants, total SOD activity was 3.2 times lower than that in the control, primarily due to a more significant decrease in the activities of Mn-SOD + Fe-SOD (6-fold) than those of Cu/Zn-SOD (2-fold) [9]. In contrast, Mn-deficient *Lupinus angustifolius* plants presented an increase in total SOD activity, attributed to an increase in Cu/Zn-SOD activity and even a 2.3-fold increase in the activity of Mn-containing isozymes [32]. Mn-deficient tomato plants also presented higher levels of mitochondrial Mn-SOD and cytosolic CuZn-SOD activities in leaves than control plants did [11]. Notably, most data on antioxidant enzyme activity under Mn deficiency conditions are from herbaceous plants, with studies on woody plants being rare. For example, in Norway spruce, Mn deficiency in needles led to a 30% increase in SOD activity compared with that in control plants [14]. In mulberry, an 8.7-fold decrease in the Mn content in leaves compared with that in the control, down to 0.27 μmol/g DW, was also accompanied by an increase in total SOD activity in leaves due to an increase in Fe-SOD activity. In this case, the activity of Mn-SOD remained unchanged, whereas the activity of Cu/Zn-SOD decreased [13].

Unlike Norway spruce, Mn-containing SOD isozymes in Scots pine needles are not detected via native PAGE [33], since their activity does not exceed 4% of the total SOD activity in needles [16]. Mn-SOD in Scots pine is known to be localized in mitochondria, and detection of its activity is possible only at certain ontogenetic stages [34]. The results of the inhibitory analysis via native PAGE indicate that the total SOD activity in the needles of the control and Mn-deficient plants was due to the activities of three Cu/Zn-SOD isozymes, which disappeared upon the addition of 5 mM H_2_O_2_ [16] or 3 mM KCN [16,35] but were not inhibited by 10 mM NaN_3_. Isozyme No. 1, characterized by the highest mobility in the polyacrylamide gel, made the greatest contribution to the total SOD activity. Analysis of the activities of the SOD isozymes on polyacrylamide gels did not reveal reliable differences between the control and Mn-deficient needles (Figure 5). On the one hand, this could be attributed to the method’s low sensitivity in comparison to spectrophotometric analysis. The absence of differences in SOD activity in the enzyme extracts before and after electrophoresis (Appendix A) excludes enzyme destruction or inactivation as a possible reason for the absence of differences between the samples on the polyacrylamide gel. Comparative analysis of SOD activities in the crude extract and in the extract purified by desalting columns with Sephadex G-25 resin (Appendix A) also confirmed the preservation of differences in SOD activity between the control and Mn-deficient needles. Since the protein loading into the loading wells corresponds to the range in which a linear relationship between protein content and signal intensity is maintained (Appendix A), we can also exclude densitometry error, which is characteristic of negatively stained bands of SOD [36]. On the other hand, the absence of differences in SOD activity on polyacrylamide gels may be explained by the unaccounted activity of SODs. For example, isozyme No. 4 in the samples when the loading wells are overloaded with protein (Figure 4A and Appendix A) also contributes to total SOD activity. In addition to the main Cu/Zn-SOD isozymes, Scots pine needles contain at least four extracellular Cu/Zn-SOD isozymes that contribute to total SOD activity. However, their amounts are below the detection limits of SOD activity after PAGE [35]. Thus, the results of the spectrophotometric analysis more closely reflect the observed changes in SOD activity. Consequently, the data obtained clearly demonstrated reduced Cu/Zn-SOD activity in the needles of Scots pine seedlings under Mn deficiency. As previously shown, the Cu and Zn contents in the needles of Mn-deficient plants did not differ from those in the control plants [15]. Therefore, we can exclude the disruption of ion homeostasis as the cause of the reduced Cu/Zn-SOD activity. Despite the widespread belief that oxidative stress develops in Mn-deficient plants [2,4], including due to a decrease in the oxygen evolution rate in chloroplasts [37], PSII inhibition can also be considered a mechanism of protection against excessive ROS generation [38,39]. This finding requires further, more detailed study, as SODs are involved in plant defense against pathogens [40,41]. Thus, the reduced level of SOD activity observed in Mn-deficient plants may be an additional factor affecting the stability of Scots pine stands under natural growing conditions in soils with Mn deficiency.

The activities of SOD, catalases, and peroxidases are believed to be functionally linked because hydrogen peroxide, which is catalyzed by SOD, is a substrate for both catalase and peroxidases [6]. Moreover, the balance between SOD and CAT or peroxidases in plant cells is crucial for determining the steady-state levels of superoxide anions and hydrogen peroxide [41]. However, a coordinated, albeit less pronounced, decrease in activity was characteristic of POD at only the 6th and 19th weeks of the experiment (Figure 3E), similar to that of SOD. Since POD is a heme-dependent oxidoreductase, iron deficiency can reduce POD activity [42]. However, under Mn deficiency conditions, the iron content in Scots pine needles actually increases [15], ruling out iron deficiency as a limiting factor for POD activity. Interestingly, in Norway spruce, Mn deficiency led to a twofold increase in POD activity in needles [43], whereas in mulberry, it had no effect on enzyme activity in leaves [13]. These data indicate a pronounced species specificity of POD activity under Mn deficiency conditions.

The lack of changes in the activities of CAT (Figure 3C) and APX (Figure 3G) in the needles of the Mn-deficient plants, in contrast to the twofold lower activity of SOD (Figure 3A), suggests that the hydrogen peroxide content formed during other biochemical reactions in the needles is at an optimal level and does not need to be neutralized. This may be one of the reasons for the absence of signs of oxidative stress in Scots pine under Mn deficiency conditions, which was determined by the lack of increases in the contents of malondialdehyde and 4-hydroxyalkenals in the roots and needles [15]. On the other hand, the lack of increase in the activities of CAT and APX in the organs of Mn-deficient plants can be explained by the absence of increased ROS generation under conditions of Mn deficiency. For example, under the toxic effect of zinc, the activity of antioxidant enzymes increases only in the roots, which are in direct contact with the metal, but not in the needles of the seedlings [30]. In the case of Mn deficiency, no changes in the activities of the studied antioxidant enzymes in plant roots were detected compared with those in the control. These findings suggest that the balance of the enzymatic antioxidant system remains intact even when root growth is inhibited under Mn deficiency conditions. This phenomenon is especially important for maintaining plant resistance to pathogenic soil microflora and root-eating insects [44].

Notably, we did not find any changes in the activity of APX, which is considered one of the key enzymes preventing the development of oxidative stress in chloroplasts [6,45]. The absence of changes in APX activity in Mn-deficient Norway spruce needles was reported by Polle et al. [14]. Some decreases in APX activity were noted in the leaves of Mn-deficient mulberry plants [13], and in tobacco, a fourfold decrease in APX activity was recorded under Mn deficiency conditions [9]. These data may indicate that APX does not play a primary role in the plant defense response under Mn deficiency conditions.

The diagnosis of Mn deficiency in plants can be challenging because of the delayed onset of symptoms [2,4]. However, by measuring chlorophyll fluorescence parameters, we can quickly identify the immediate response to Mn deficiency [4]. Our research revealed a decrease in SOD activity in the needles of Mn-deficient plants, suggesting that this parameter can be used to diagnose Mn deficiency in Scots pine.

## 5. Conclusions

The experimental data we obtained refute our hypothesis that Mn deficiency should not cause a decrease in total SOD activity. Inhibitory analysis via PAGE indicated that total SOD activity was determined by the activity of the Cu/Zn-containing SOD isozymes. This significantly distinguishes Scots pine from other plants, including Norway spruce, which typically utilize Mn- and/or Fe-containing SOD isozymes. The predominance of Cu/Zn-containing SOD isozymes in Scots pine should ensure the stable functioning of this enzyme under conditions of Mn deficiency. However, with Mn deficiency in needles, regardless of the duration of exposure and the severity of this deficiency, a reduced level of Cu/Zn-SOD activity was observed. Remarkably, this occurred even when plant growth was not disturbed. Despite significant differences in the Mn content in plant organs, a lack of coordinated increase in the activity of antioxidant enzymes was observed. This pattern of antioxidant enzyme activity may indicate that the inhibition of PSII in Scots pine under Mn deficiency conditions is a protective strategy that prevents ROS formation in needles. Furthermore, maintaining the balance of the enzymatic antioxidant system in the roots of pine seedlings despite growth inhibition may be necessary for plant resistance to soil pathogens and insect pests.

## Figures and Tables

**Figure 1 biology-15-00101-f001:**
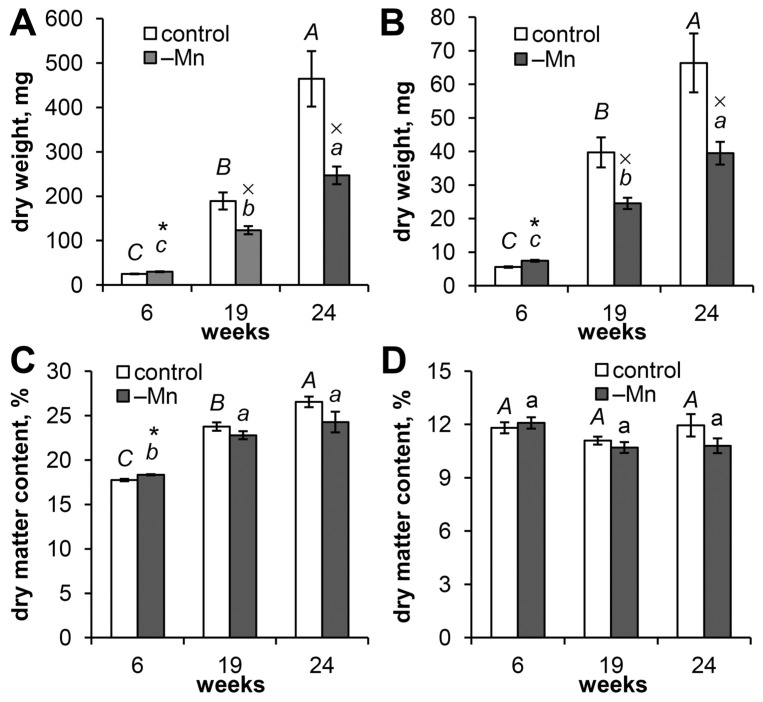
Dry weight (**A**,**B**) and dry matter content (**C**,**D**) of the organs of Scots pine seedlings: needles (**A**,**C**) and roots (**B**,**D**). The mean values ± SEs are given (*n* = 6–12). Different letters (capital for the control and lowercase for Mn-deficient) indicate significant differences between time points (*p* < 0.05) according to ANOVA followed by Duncan’s post hoc test (regular letters) or by Student–Newman–Keuls post hoc test (italic letters). Pairwise comparisons of the means with controls at corresponding time points were performed using Student’s *t*-test for normally distributed data (significant differences at *p* < 0.05 denoted by asterisks (*)) or the Mann–Whitney rank sum test when the *t*-test was not applicable (significant differences at *p* < 0.05 denoted by multiplication symbols (×)).

**Figure 2 biology-15-00101-f002:**
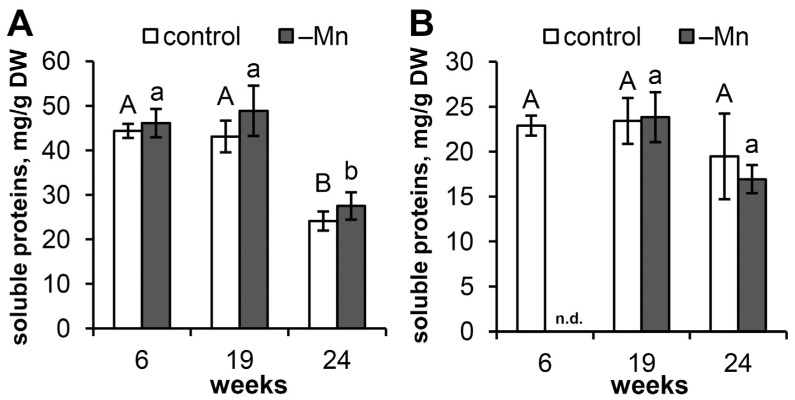
Soluble protein content in the organs of Scots pine seedlings: needles (**A**) and roots (**B**). The mean values ± SEs are given (*n* = 4–6, n.d.—no data). Different letters (capital for the control and lowercase for Mn-deficient) indicate significant differences between time points (*p* < 0.05) according to ANOVA followed by Duncan’s post hoc test.

**Figure 3 biology-15-00101-f003:**
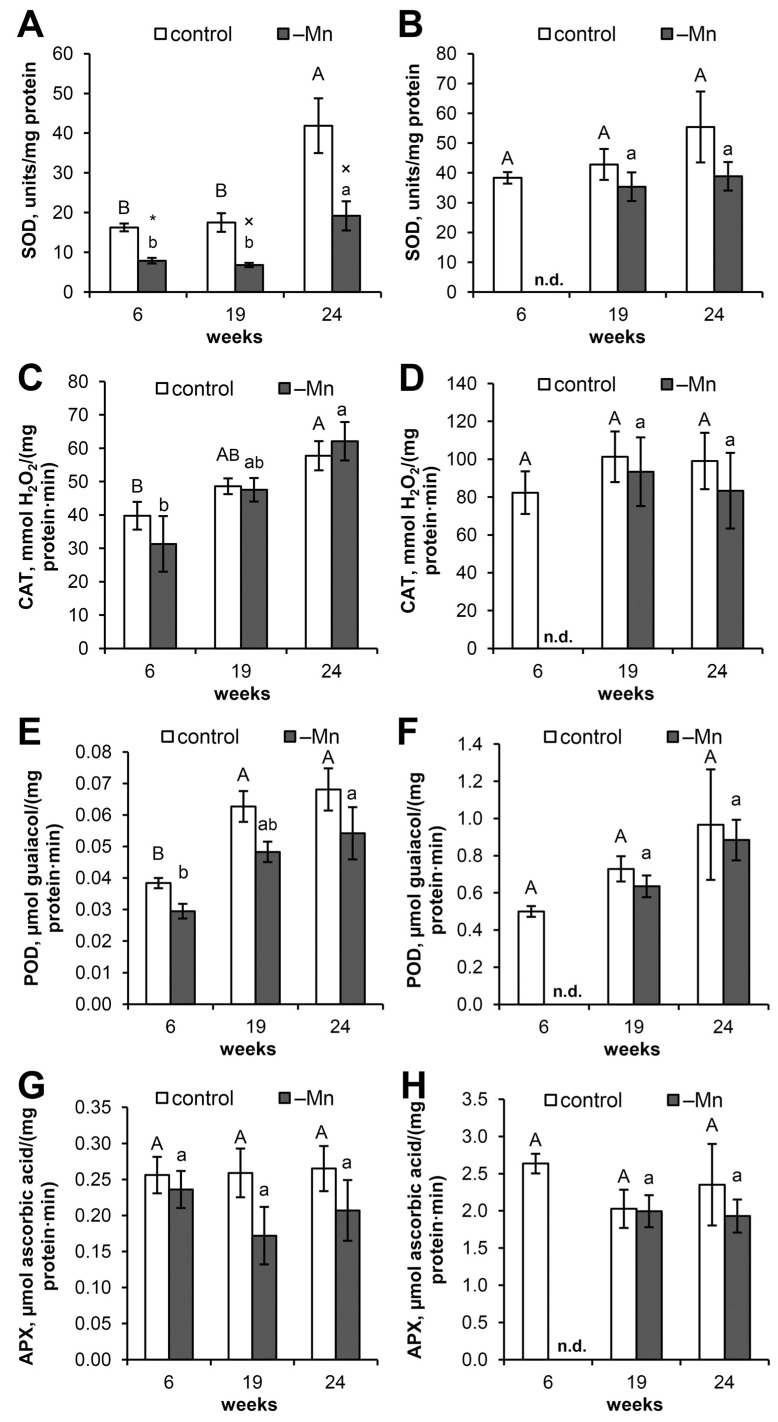
Activity of the following antioxidant enzymes: (**A**,**B**) superoxide dismutase (SOD); (**C**,**D**) catalase (CAT); (**E**,**F**) guaiacol peroxidase (POD); and (**G**,**H**) ascorbate peroxidase (APX) in the needles (**A**,**C**,**E**,**G**) and roots (**B**,**D**,**F**,**H**) of Scots pine seedlings. Enzyme activities are expressed per mg of soluble protein. The mean values ± SEs are given (*n* = 4–6, n.d.—no data). Different letters (capital for the control and lowercase for Mn-deficient) indicate significant differences between time points (*p* < 0.05) according to ANOVA followed by Duncan’s post hoc test (regular letters). Pairwise comparisons of the means with controls at corresponding time points were performed using Student’s *t*-test for normally distributed data (significant differences at *p* < 0.05 denoted by asterisks (*)) or the Mann–Whitney rank sum test when the *t*-test was not applicable (significant differences at *p* < 0.05 denoted by multiplication symbols (×)).

**Figure 4 biology-15-00101-f004:**
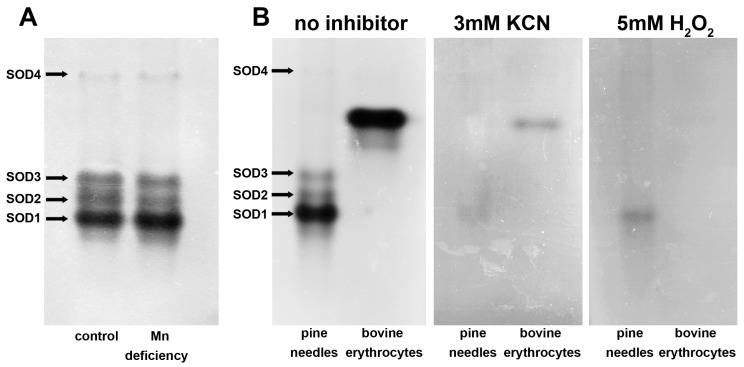
Activity staining of SOD from Scots pine needles and SOD from bovine erythrocytes after native PAGE: (**A**) SOD from pine needles (80 µg of protein per loading well); (**B**) results of inhibitory analysis of SOD in the presence of 3 mM KCN or in the presence of 5 mM H_2_O_2_. During processing, the original images of the gels were converted to black and white and inverted for easier perception.

**Figure 5 biology-15-00101-f005:**
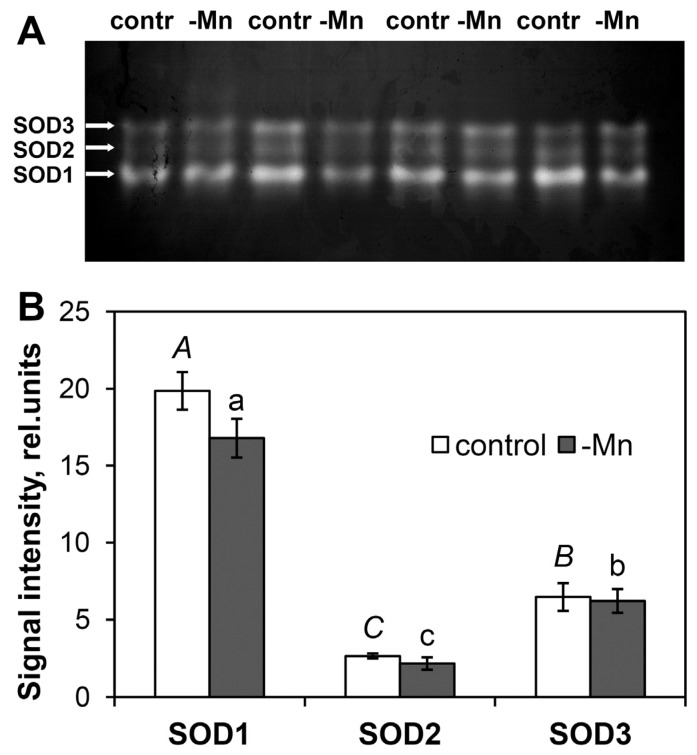
Comparative analysis of SOD activity in the needles of Scots pine seedlings at the 24th week of the experiment after native PAGE: (**A**) typical polyacrylamide gel stained after gel electrophoresis; (**B**) results of quantitative assessment of SOD isozyme activities performed in Image Studio Digits. Different letters (capital for the control and lowercase for Mn-deficient) indicate significant differences between activities of SOD isozymes (*p* < 0.05) according to ANOVA followed by Duncan’s post hoc test (regular letters) or by Student–Newman–Keuls post hoc test (italic letters). There were no statistically significant differences in the activities of SOD isozymes between the control and Mn-deficient plants.

**Table 1 biology-15-00101-t001:** Manganese content in the roots and needles of Scots pine seedlings.

Organ	6th Week		19th Week		24th Week	
	Control	Mn-Deficient	Control	Mn-Deficient	Control	Mn-Deficient
Roots *	3.17 ± 0.34 ***A***	0.04 ± 0.01 b, ×	0.41 ± 0.02 ***B***	0.10 ± 0.01 a, ×	0.71 ± 0.05 ***AB***	0.10 ± 0.00 a, ×
Needles *	5.75 ± 0.21 A	0.34 ± 0.03 *a*, ×	6.18 ± 0.63 A	0.11 ± 0.01 *b*, ×	5.92 ± 0.55 A	0.10 ± 0.02 *b*, ×

* Data adapted from [15]. The mean values ± SEs are given (*n* = 6–12). Different letters (capital letters for control and lowercase letters for Mn-deficient) indicate significant differences between time points (*p* < 0.05) according to ANOVA followed by Dunn’s post hoc test (bold italic letters), or by Student–Newman–Keuls post hoc test (italic letters), or by Duncan’s post hoc test (regular letters). Pairwise comparisons of the means with controls at corresponding time points were performed using the Mann–Whitney rank sum test (significant differences at *p* < 0.05 are denoted by multiplication symbols (×)).

## Data Availability

The datasets generated and/or analyzed during the current study are available from the corresponding authors upon reasonable request.

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
