# Peer review of "Enzymatic Antioxidant Defense System of Scots Pine Seedlings Under Conditions of Progressive Manganese Deficiency"

_biology, 2026, doi:10.3390/biology15010101_

Round 1
Reviewer 1 Report
Comments and Suggestions for Authors
The work is scientifically sound and methodologically robust overall, addressing a clear knowledge gap, particularly with respect to woody plants, which are underrepresented in Mn nutrition and oxidative stress studies. The manuscript is suitable for publication in Biology after minor revisions, primarily aimed at enhancing clarity, methodological transparency, and figure presentation.
The authors repeatedly refer to the absence of oxidative stress based on previous work and indirect indicators. While this is acceptable, the manuscript would benefit from a clearer statement in the Discussion explicitly acknowledging that ROS levels were not directly quantified in the present study, and that conclusions regarding oxidative stress are inferred from enzyme activity patterns and earlier data.
Native PAGE analysis shows no statistically significant differences in individual SOD isozyme activities, despite a clear reduction in total SOD activity measured spectrophotometrically. This apparent discrepancy is addressed, but the explanation could be strengthened by:
-
More explicitly discussing methodological sensitivity limits of native PAGE, and
-
Clarifying the potential contribution of extracellular Cu/Zn-SOD, which is not fully resolved by the current approach
The Discussion would benefit from a more explicit mechanistic hypothesis explaining why Mn deficiency selectively affects Cu/Zn-SOD activity, despite Cu/Zn-SOD not being Mn-dependent. Possible indirect effects (e.g., altered redox signalling, transcriptional regulation, or apoplastic processes) could be more clearly framed as working hypotheses.
-
Please ensure consistent capitalisation of enzyme names (e.g., “ascorbate peroxidase” vs. “Ascorbate Peroxidase”).
-
Specify more clearly whether enzyme activities are expressed per mg soluble protein or per mg total protein throughout the manuscript.
-
Figure 3 is information-rich but visually dense. Consider improving readability by slightly increasing font size and/or improving contrast between control and Mn-deficient bars.
-
Ensure that all abbreviations used in figures are defined either in the figure caption or consistently in the main text.
Comments on the Quality of English Language
The English is generally clear, but minor grammatical and stylistic issues remain (e.g., occasional long sentences and repetition in the Discussion). A light language edit would improve readability without altering scientific content
Author Response
Comments 1: The work is scientifically sound and methodologically robust overall, addressing a clear knowledge gap, particularly with respect to woody plants, which are underrepresented in Mn nutrition and oxidative stress studies. The manuscript is suitable for publication in Biology after minor revisions, primarily aimed at enhancing clarity, methodological transparency, and figure presentation.
Response 1: We thank the esteemed reviewer for his positive assessment of our work and suggestions for revision of the manuscript.
Comments 2: The authors repeatedly refer to the absence of oxidative stress based on previous work and indirect indicators. While this is acceptable, the manuscript would benefit from a clearer statement in the Discussion explicitly acknowledging that ROS levels were not directly quantified in the present study, and that conclusions regarding oxidative stress are inferred from enzyme activity patterns and earlier data.
Response 2: Indeed, our assessment of the absence of oxidative damage in Scots pine seedling organs was based on previously obtained data on the content of lipid peroxidation products in the roots and needles of the plants. To avoid misinterpretation of this statement, we have amended the text in accordance with the reviewer's recommendation.
Comments 3: Native PAGE analysis shows no statistically significant differences in individual SOD isozyme activities, despite a clear reduction in total SOD activity measured spectrophotometrically. This apparent discrepancy is addressed, but the explanation could be strengthened by:
- More explicitly discussing methodological sensitivity limits of native PAGE, and
- Clarifying the potential contribution of extracellular Cu/Zn-SOD, which is not fully resolved by the current approach
Response 3: In accordance with the reviewer's recommendations, we expanded the Discussion section regarding the sensitivity of SOD activity analysis via PAGE.
Comments 4: he Discussion would benefit from a more explicit mechanistic hypothesis explaining why Mn deficiency selectively affects Cu/Zn-SOD activity, despite Cu/Zn-SOD not being Mn-dependent. Possible indirect effects (e.g., altered redox signalling, transcriptional regulation, or apoplastic processes) could be more clearly framed as working hypotheses.
Response 4: In accordance with the reviewer's recommendations, we have expanded the Discussion regarding the effect of manganese deficiency on Cu/Zn-SOD activity.
Comments 5: Please ensure consistent capitalisation of enzyme names (e.g., “ascorbate peroxidase” vs. “Ascorbate Peroxidase”).
Response 5: We used enzyme names in accordance with the recommendations of the International Union of Biochemistry and Molecular Biology. As a rule, the names of well-known enzymes are written with lowercase letters. See: Cornish-Bowden, A., 2014. Current IUBMB recommendations on enzyme nomenclature and kinetics. Perspectives in Science, 1(1-6), pp.74-87., or https://iubmb.qmul.ac.uk/enzyme/EC1/11/1/11.html. We used capitalized enzyme names in the manuscript subsection titles in accordance with the rules for formatting subsection headings.
Comments 6: Specify more clearly whether enzyme activities are expressed per mg soluble protein or per mg total protein throughout the manuscript.
Response 6: The activities of all the enzymes studied are expressed per mg of soluble protein. The necessary clarifications have been added to the manuscript.
Comments 7: Figure 3 is information-rich but visually dense. Consider improving readability by slightly increasing font size and/or improving contrast between control and Mn-deficient bars.
Response 7: Figure 3 has been corrected according to the reviewer's recommendations.
Comments 8: Ensure that all abbreviations used in figures are defined either in the figure caption or consistently in the main text.
Response 8: In accordance with the reviewer's recommendation, abbreviations have been added to the figure captions.
Comments 9: Comments on the Quality of English Language
The English is generally clear, but minor grammatical and stylistic issues remain (e.g., occasional long sentences and repetition in the Discussion). A light language edit would improve readability without altering scientific content
Response 9: In accordance with the reviewer's recommendation, English editing was performed on the revised version of the manuscript.
Reviewer 2 Report
Comments and Suggestions for Authors
General Comments
The manuscript addresses an important topic in plant stress physiology by examining the role of enzymatic antioxidant defense mechanisms under progressive manganese (Mn) deficiency in Scots pine seedlings. Manganese is a key micronutrient, particularly for photosystem II function, and its deficiency is known to induce oxidative stress. The study is relevant and timely, especially for understanding micronutrient stress responses in forest tree species. Overall, the abstract is clearly written, and the study objectives are well aligned with the experimental outcomes. However, some aspects require clarification and strengthening to improve scientific rigor and impact.
Strengths
-
Relevance of the Study: The focus on Mn deficiency and its link to oxidative stress and antioxidant enzymes in a forest species is scientifically valuable and ecologically significant.
-
Clear Objective: The study clearly investigates the response of key antioxidant enzymes (SOD, CAT, APX, and POD) under progressive Mn deficiency.
-
Organ-Specific Analysis: The comparison between needles and roots provides useful insights into tissue-specific responses.
-
Novel Observation: The finding that SOD activity decreased rather than increased under Mn deficiency, even without growth reduction, is interesting and challenges commonly held assumptions.
Weaknesses and Points for Improvement
-
Mechanistic Explanation: While the abstract reports reduced SOD activity under Mn deficiency, the physiological or molecular basis for this decrease—especially the exclusive involvement of Cu/Zn-SOD—is not sufficiently explained.
-
Oxidative Stress Indicators: The abstract does not mention direct measurements of oxidative damage (e.g., Hâ‚‚Oâ‚‚ content, lipid peroxidation), which would strengthen the link between Mn deficiency and oxidative stress.
-
Severity Definition: The term “progressive Mn deficiency” is used, but the levels or stages of deficiency are not specified in the abstract. Including this information would improve clarity.
-
Root Response Discussion: The lack of antioxidant enzyme response in roots is mentioned but not interpreted, which limits understanding of whole-plant adaptation strategies.
-
Broader Implications: The ecological or forestry-related implications of reduced antioxidant capacity under Mn deficiency could be emphasized more clearly.
Specific Comments on the Abstract
-
The abstract is well structured and concise but would benefit from a clearer statement of the hypothesis.
-
The conclusion could be strengthened by explicitly stating how these findings advance current understanding of Mn deficiency responses in coniferous species.
Recommendation
Minor Revision
The manuscript presents valuable and well-organized findings, but minor revisions are recommended to improve clarity, mechanistic interpretation, and contextual relevance. Addressing the points above will enhance the overall quality and impact of the study.
Author Response
Comments 1: General Comments
The manuscript addresses an important topic in plant stress physiology by examining the role of enzymatic antioxidant defense mechanisms under progressive manganese (Mn) deficiency in Scots pine seedlings. Manganese is a key micronutrient, particularly for photosystem II function, and its deficiency is known to induce oxidative stress. The study is relevant and timely, especially for understanding micronutrient stress responses in forest tree species. Overall, the abstract is clearly written, and the study objectives are well aligned with the experimental outcomes. However, some aspects require clarification and strengthening to improve scientific rigor and impact.
Response 1: We are grateful to the distinguished reviewer for his careful analysis of our manuscript and suggestions for its revision.
Comments 2: Strengths
Relevance of the Study: The focus on Mn deficiency and its link to oxidative stress and antioxidant enzymes in a forest species is scientifically valuable and ecologically significant.
Clear Objective: The study clearly investigates the response of key antioxidant enzymes (SOD, CAT, APX, and POD) under progressive Mn deficiency.
Organ-Specific Analysis: The comparison between needles and roots provides useful insights into tissue-specific responses.
Novel Observation: The finding that SOD activity decreased rather than increased under Mn deficiency, even without growth reduction, is interesting and challenges commonly held assumptions.
Response 2: We are grateful to the reviewer for the positive assessment of our work.
Comments 3: Weaknesses and Points for Improvement
Mechanistic Explanation: While the abstract reports reduced SOD activity under Mn deficiency, the physiological or molecular basis for this decrease—especially the exclusive involvement of Cu/Zn-SOD—is not sufficiently explained.
Response 3: We have added a description of the possible reasons for the observed reduced levels of Cu/Zn-SOD activity in Mn-deficient plants in the Discussion.
Comments 4: Oxidative Stress Indicators: The abstract does not mention direct measurements of oxidative damage (e.g., H2O2 content, lipid peroxidation), which would strengthen the link between Mn deficiency and oxidative stress.
Response 4: Indeed, we did not measure the levels of reactive oxygen species in plant organs during this experiment. The conclusion regarding the absence of oxidative stress in seedling organs is based on the results of measuring lipid peroxidation products—malondialdehyde and 4-hydroxyalkenals. Since these data have already been published, we cite them in the text of this manuscript. However, in the revised version of the manuscript, we have provided a clearer justification for the data underlying the absence of oxidative stress.
Comments 5: Severity Definition: The term “progressive Mn deficiency” is used, but the levels or stages of deficiency are not specified in the abstract. Including this information would improve clarity.
Response 5: Unfortunately, the structure and length of the Abstract section are strictly regulated by the journal's guidelines, preventing a more detailed description of the work performed and the results obtained. However, taking into account the reviewer's recommendations, we have attempted to modify the section by adding the necessary information.
Comments 6: Root Response Discussion: The lack of antioxidant enzyme response in roots is mentioned but not interpreted, which limits understanding of whole-plant adaptation strategies.
Response 6: As recommended by the reviewer, Root Response Discussion has been added to the text of the revised manuscript.
Comments 7: Broader Implications: The ecological or forestry-related implications of reduced antioxidant capacity under Mn deficiency could be emphasized more clearly.
Response 7: We expanded the description of the possible ecological consequences of the observed decrease in SOD activity in Scots pine needles.
Comments 8: Specific Comments on the Abstract. The abstract is well structured and concise but would benefit from a clearer statement of the hypothesis.
Response 8: In accordance with the reviewer's recommendation, a hypothesis has been formulated in the Abstract section of the revised version of the manuscript.
Comments 9: The conclusion could be strengthened by explicitly stating how these findings advance current understanding of Mn deficiency responses in coniferous species.
Response 9: The Conclusion section has been modified according to the reviewer's recommendations.
Comments 10: Recommendation. Minor Revision
The manuscript presents valuable and well-organized findings, but minor revisions are recommended to improve clarity, mechanistic interpretation, and contextual relevance. Addressing the points above will enhance the overall quality and impact of the study.
Response 11: We thank the reviewer for his comments and suggestions, all of which were taken into account when revising the manuscript.
Reviewer 3 Report
Comments and Suggestions for Authors
The manuscript (MS) entitled “Enzymatic antioxidant defence system of Scots pine seedlings under conditions of progressive manganese deficiency”, is important for this species. This study examines the changes in SOD, ascorbate peroxidase, catalase, and guaiacol peroxidase activities in Scots pine seedling organs over an extended period (up to 24 weeks) of Mn deficiency; however, the MS has omissions:
-In this study, it is important to know if there were variations in the developmental variables. The author should include the developmental variables of the seedlings.: seedling length, number of roots, number of needles, fascicles among others (at least during the 24th week).
- In addition, the author must include the dry matter. Dry matter (DM) contents is estimated using the following formula: DM = (dry weight/fresh weight) × 100.
-In Figure 5B. Highlighting the differences between treatments.
-Discussion part. It is not described what the authors consider to be the novelty of the presented work and its practical significance.
Author Response
Comments 1: The manuscript (MS) entitled “Enzymatic antioxidant defence system of Scots pine seedlings under conditions of progressive manganese deficiency”, is important for this species. This study examines the changes in SOD, ascorbate peroxidase, catalase, and guaiacol peroxidase activities in Scots pine seedling organs over an extended period (up to 24 weeks) of Mn deficiency; however, the MS has omissions:
-In this study, it is important to know if there were variations in the developmental variables. The author should include the developmental variables of the seedlings.: seedling length, number of roots, number of needles, fascicles among others (at least during the 24th week).
Response 1: Unfortunately, we are unable to include the data requested by the reviewer, as some of it has already been published. Furthermore, counting the number of roots or needles in 19-week-old plants, especially 24-week-old plants is an extremely labor-intensive task, which we did not aim to solve in this experiment. By the 10th week of seedling development, the number of needles reached 90, and the taproot system of seedlings contained up to 110 first-order lateral roots and approximately 80 second-order lateral roots (see Plants 2024, 13, 2227). Unfortunately, we do not have photographs of the plants from the current experiment, but an overview of 18-19-week-old Scots pine plants can be obtained from the images presented in Zlobin et al., 2019 (Photosynthesis Research, 139:307–323). In this context, the most reliable indicator of plant development is weight. However, we would like to emphasize that no visually discernible differences were observed between the Mn-deficient plants and the control plants until the 22nd week of the experiment. During this period, the plants began to show signs of chlorosis.
Comments 2: In addition, the author must include the dry matter. Dry matter (DM) contents is estimated using the following formula: DM = (dry weight/fresh weight) × 100.
Response 2: In accordance with the reviewer's recommendation, data on the dry matter content of roots and needles have been added to the revised version of the manuscript.
Comments 3: In Figure 5B. Highlighting the differences between treatments.
Response 3: Figure 5B shows no statistically significant differences between the control and Mn-deficient plants. Since the current version of the figure raised several questions, we have modified the figure and added clarifications to the caption.
Comments 4: Discussion part. It is not described what the authors consider to be the novelty of the presented work and its practical significance.
Response 4: The Discussion section has been revised in accordance with the reviewer's recommendations.
Reviewer 4 Report
Comments and Suggestions for Authors
The manuscript explains “Enzymatic antioxidant defence system of Scots pine seedlings under conditions of progressive manganese deficiency”. After a careful review, I found this work is interesting and worth publishing in the Biology. Therefore, I propose this work can be published after major revision as following:
- In abstract, the authors should mention the treatments of Mn for Scots pine seedling for better understanding.
- Line 34: In abstract, the authors mentioned the plant organs but did not specify the organ name.
- Line33-43: In abstract, the authors should explain all organs response to Mn deficiency and then overall impact of Mn on Scots pine seedlings.
- Line 84: In introduction, the authors should use abbreviation after first explained in the beginning.
- Line 90 and 373: In introduction, the authors should check “Manganese deficiency per se should…SOD activity”. In this sentence, what is “per se”?
- Line 93: In introduction, the authors should specify the plant organs or write different organs.
- Line 98-100: In material and methodology, the authors should revise this sentence and write it in grammatically correct English.
- Line 100-101: In material and methodology, the cited reference used 5μM MnSO4. Why the authors used 5.2 μM Mn? What is reason to reduce the concentration 0263 μM (26.3 nM) Mn? Which source did the authors use for Mn? Why authors did not use deificient without any addition of Mn in hydro-culture?
- In material and methodology, the authors should mention the treatment application and change of hydroponic solution periodically for whole growing season.
- Line 102: In material and methodology, the authors only collected data at 6, 19, 24 weeks. What is the reason behind it and write it in the material and methodology? If there is specific growth changes in this time then should also mention it.
- Line 104: In material and methodology, the authors mentioned 1 or 2 seedlings to make composite sampling but 1 or 2 samples are not enough for composite sampling. Did the authors use the replication of each time sampling?
- Line 125: In material and methodology, the authors can remove “+ sign” from +4°C.
- Line 126: In material and methodology, the authors should mention specific enzyme activity and give the reference of this methodology.
- Line 195-196: In material and methodology, the authors should specify the plant organs or write different organs.
- Line 210: In material and methodology, what is mean by totals presented? The authors can change total results presented…standard errors.
- In material and methodology, the authors should presented the fresh weight and dry weight as well along with roots and needles.
- Line 216: In results section, why was the Mn concents remain constant/stabilized at 19th and 24th week? Are the authors not changing hydroponic culture periodically?
- Line 222 and 242: In results section, the authors are mentioning n=6-12 while in material and methodology it is different. Kindly correct it in whole manuscript.
- Line 251: In results section, Are the authors comparing needles Mn deficient with control needle or whole plant? Kindly write it clearly in whole manuscript.
- Line 282 and 286: : In results section, the authors should check the figure number. It is figure 3E and 3F instead of figure 2F and 2E.
- The authors should provide more clear figure of 4 for PAGE.
- In discussion, the authors should give logical discussion and reasons along previous findings and trying to add more details about reasoning about Mn deficiency in Scots pine seedling especially different organs.
- In conclusion, the authors should give more structurally organized conclusion from their findings to their suggestions
- Comprehensive English language corrections are required in the whole manuscript.
Author Response
Comments 1: The manuscript explains “Enzymatic antioxidant defence system of Scots pine seedlings under conditions of progressive manganese deficiency”. After a careful review, I found this work is interesting and worth publishing in the Biology. Therefore, I propose this work can be published after major revision as following:
Response 1: We are grateful to the distinguished reviewer for his careful review of our manuscript and detailed comments for its improvement.
Comments 2: In abstract, the authors should mention the treatments of Mn for Scots pine seedling for better understanding.
Response 2: We provided a detailed description of the experimental design in the Materials and Methods section. Unfortunately, space limitations in the Abstract do not allow for a detailed description of the experiment in this section.
Comments 3: Line 34: In abstract, the authors mentioned the plant organs but did not specify the organ name.
Response 3: In the revised version of the manuscript, we indicated that we are referring to the roots and needles of Scots pine seedlings.
Comments 4: Line33-43: In abstract, the authors should explain all organs response to Mn deficiency and then overall impact of Mn on Scots pine seedlings.
Response 4: In accordance with the reviewer's recommendation, appropriate edits have been made.
Comments 5: Line 84: In introduction, the authors should use abbreviation after first explained in the beginning.
Response 5: In accordance with the reviewer's recommendations, appropriate changes have been made to the text of the manuscript.
Comments 6: Line 90 and 373: In introduction, the authors should check “Manganese deficiency per se should…SOD activity”. In this sentence, what is “per se”?
Response 6: “per se” is a Latin phrase meaning “by itself, in itself, or of itself”. This term is frequently used in scientific literature. In the original version of this manuscript, this term was not italicized, which may have caused confusion. We apologize for this.
Comments 7: Line 93: In introduction, the authors should specify the plant organs or write different organs.
Response 7: We have added the necessary clarifications to the manuscript.
Comments 8: Line 98-100: In material and methodology, the authors should revise this sentence and write it in grammatically correct English.
Response 8: The English language of the specified sentence has been edited.
Comments 9: Line 100-101: In material and methodology, the cited reference used 5μM MnSO4. Why the authors used 5.2 μM Mn? What is reason to reduce the concentration 0263 μM (26.3 nM) Mn? Which source did the authors use for Mn? Why authors did not use deificient without any addition of Mn in hydro-culture?
Response 9: We have added the necessary information to the description of the experimental design.
Comments 10: In material and methodology, the authors should mention the treatment application and change of hydroponic solution periodically for whole growing season.
Response 10: We have added the necessary information to the description of the experimental design.
Comments 11: Line 102: In material and methodology, the authors only collected data at 6, 19, 24 weeks. What is the reason behind it and write it in the material and methodology? If there is specific growth changes in this time then should also mention it.
Response 11: We have added the necessary information to the description of the experimental design.
Comments 12: Line 104: In material and methodology, the authors mentioned 1 or 2 seedlings to make composite sampling but 1 or 2 samples are not enough for composite sampling. Did the authors use the replication of each time sampling?
Response 12: We apologize for using incorrect terminology. By "composite sample," we meant a sample consisting of several individual plants. Owing to the small plant weights in the early stages of the experiment, it was necessary to obtain an average sample weight of 150-200 mg FW. As the plant weight subsequently increased, the number of plants combined decreased. In the revised version of the manuscript, we use the term "combined sample."
Comments 13: Line 125: In material and methodology, the authors can remove “+ sign” from +4°C.
Response 13: In accordance with the reviewer's recommendation, the “+” sign has been removed from the revised version of the manuscript.
Comments 14: Line 126: In material and methodology, the authors should mention specific enzyme activity and give the reference of this methodology.
Response 14: The necessary references have been added to the revised version of the manuscript.
Comments 15: Line 195-196: In material and methodology, the authors should specify the plant organs or write different organs.
Response 15: In accordance with the reviewer's recommendation, the description of this text has been changed.
Comments 16: Line 210: In material and methodology, what is mean by totals presented? The authors can change total results presented…standard errors.
Response 16: We replaced the term "totals" with "data." We would like to emphasize that the statistical analysis was performed on all the data collected. No procedures involving the exclusion of any results were applied. Therefore, the mean values and their standard errors are presented in the article "as is."
Comments 17: In material and methodology, the authors should presented the fresh weight and dry weight as well along with roots and needles.
Response 17: We have added the necessary information to the description in the Materials and Methods section.
Comments 18: Line 216: In results section, why was the Mn concents remain constant/stabilized at 19th and 24th week? Are the authors not changing hydroponic culture periodically?
Response 18: We have added the necessary information to the description of the experimental design.
Comments 19: Line 222 and 242: In results section, the authors are mentioning n=6-12 while in material and methodology it is different. Kindly correct it in whole manuscript.
Response 19: In accordance with the reviewer's recommendation, we clarified the description of the number of replicates.
Comments 20: Line 251: In results section, Are the authors comparing needles Mn deficient with control needle or whole plant? Kindly write it clearly in whole manuscript.
Response 20: All the physiological parameters discussed in this manuscript were obtained for the roots or needles of the seedlings. Comparisons between "control plants" and "Mn-deficient plants" are comparisons between plant groups.
Comments 21: Line 282 and 286: : In results section, the authors should check the figure number. It is figure 3E and 3F instead of figure 2F and 2E.
Response 21: Indeed, there was an error in the figure references. Thank you for the tip. The necessary corrections have been made in the revised manuscript.
Comments 22: The authors should provide more clear figure of 4 for PAGE.
Response 22: In accordance with the reviewer's recommendation, the presentation of Figure 4 was changed in the revised version of the manuscript.
Comments 23: In discussion, the authors should give logical discussion and reasons along previous findings and trying to add more details about reasoning about Mn deficiency in Scots pine seedling especially different organs.
Response 23: We have modified the Discussion section on the basis of the reviewer's recommendation.
Comments 24: In conclusion, the authors should give more structurally organized conclusion from their findings to their suggestions
Response 24: The Conclusion section has been modified according to the reviewer's recommendations.
Comments 25: Comprehensive English language corrections are required in the whole manuscript.
Response 25: In accordance with the reviewer's recommendation, English editing was performed on the revised version of the manuscript.
Round 2
Reviewer 4 Report
Comments and Suggestions for Authors
The authors have addressed most of my comments and suggestions, but the following queries should still be addressed in the manuscript:
- In abstract, the author mentioned “Mn-deficient plants increased from 17-fold at the beginning of the experiment to 59-fold at the end.” How could it be increased from 17 fold to 59 fold at the end if there is without added Mn?
- In material and methodology, the authors are contradicting their statements, first the authors said Mn-deficient (without added Mn), and later 26.3 nM for the Mn-deficient group? The other is that the authors used 5 μM MnSOâ‚„ but mentioned 5.2 μM Mn. Are they not using ddHâ‚‚O for the solution preparation?
- In material and methodology, the authors mentioned 1 or 2 seedlings were grouped into combined samples and separated in the same manner. This sentence is still confusing; kindly make it clear for better understanding of the readers.
Author Response
We thank the esteemed reviewer for reanalyzing the revised version of our manuscript.
Comments 1: In abstract, the author mentioned “Mn-deficient plants increased from 17-fold at the beginning of the experiment to 59-fold at the end.” How could it be increased from 17 fold to 59 fold at the end if there is without added Mn?
Response 1: In response to one of the reviewer's comments, we have added detailed information to the manuscript's abstract describing the progressive Mn deficiency. We would like to point out to the reviewer that only a partial sentence was included in the comment. The full sentence, "(the difference in Mn content between the needles of control and Mn-deficient plants increased from 17-fold at the beginning of the experiment to 59-fold at the end)" indicates an increase in the differences in Mn content between control and Mn-deficient plants. Therefore, we do not see any contradiction here.
Comments 2: In material and methodology, the authors are contradicting their statements, first the authors said Mn-deficient (without added Mn), and later 26.3 nM for the Mn-deficient group? The other is that the authors used 5 μM MnSOâ‚„ but mentioned 5.2 μM Mn. Are they not using ddHâ‚‚O for the solution preparation?
Response 2: When the composition of the growth medium used for growing plants is specified, nominal concentrations of elements are provided. However, the actual concentrations in the growth medium may differ from the nominal concentrations for various reasons. Therefore, it is customary during the experiment to monitor the actual concentrations of the target elements and provide these data. The Mn concentration in the growth medium of the control plants during the experiment ranged from 5.21±0.33 μM. These values are considered to be as close as possible to the nominal concentrations. Throughout the experiment, we used ddH2O to prepare the nutrient stocks and growth medium, maintaining the utmost purity of the materials and equipment used. However, Mn is present as an admixture in mineral salts and even in the air, which can lead to its detection in nutrient solutions using high-precision analytical methods. The fluctuations in Mn concentrations in the nutrient solution used to grow the Mn-deficient plants did not exceed 26.3 ± 2.2 nM, indicating that the differences between the control and Mn-deficient groups averaged approximately 200-fold. These differences were discussed in our previous work, which is referenced below. In the revised version of the manuscript, we have added the standard errors characterizing the range of actual Mn concentrations in the nutrient solutions used.
Comments 3: In material and methodology, the authors mentioned 1 or 2 seedlings were grouped into combined samples and separated in the same manner. This sentence is still confusing; kindly make it clear for better understanding of the readers.
Response 3: We apologize for the inadequacy of our previous edits. This section has been revised in the updated version of the manuscript. We believe that this approach will be more comprehensible to readers.